# Identification and Functional Prediction of CircRNAs in Leaves of F1 Hybrid Poplars with Different Growth Potential and Their Parents

**DOI:** 10.3390/ijms24032284

**Published:** 2023-01-23

**Authors:** Weixi Zhang, Zhengsai Yuan, Jing Zhang, Xiaohua Su, Qinjun Huang, Qi Liu, Changjun Ding

**Affiliations:** 1State Key Laboratory of Tree Genetics and Breeding, Research Institute of Forestry, Chinese Academy of Forestry, Beijing 100091, China; 2State Key Laboratory of Tree Breeding and Cultivation of State Forestry Administration, Research Institute of Forestry, Chinese Academy of Forestry, Beijing 100091, China; 3Co-Innovation Center for Sustainable Forestry in Southern China, Nanjing Forestry University, Nanjing 210037, China

**Keywords:** *Populus deltoides*, CircRNA, heterosis, non-additive expression, single-parent expression, differential expression, WGCNA

## Abstract

Circular RNAs (CircRNAs) regulate plant growth and development; however, their role in poplar heterosis is unclear. We identified 3722 circRNAs in poplar leaves, most of which were intergenic (57.2%) and exonic (40.2%). The expression of circRNAs in F1 hybrids with high growth potential was higher than that in those with low growth potential. Non-additive expression of circRNAs and single-parent expression of circRNAs (SPE-circRNAs) might regulate poplar heterosis through microRNA sponging and protein translation, respectively. DECs among F1 hybrids with different growth potentials might regulate the growth potential of poplar via microRNA sponging. Correlation analysis between circRNA expression and its parent gene expression showed that SPE-M circRNA (circRNAs expressed by male parent only) might regulate poplar heterosis by inhibiting parent gene expression, while other circRNAs might regulate poplar heterosis by enhancing parent gene expression. Weighted correlation network analysis of gene/circRNA expression showed that circRNAs mainly regulate poplar heterosis via carbohydrate metabolism, amino acid metabolism, energy metabolism, and material transport. In addition, we identified seven circRNAs that positively or negatively regulate poplar heterosis. Thus, non-additively expressed circRNAs and SPE circRNAs are involved in regulating poplar heterosis, and DECs among F1 hybrids with different growth potentials were involved in regulating poplar growth potential.

## 1. Introduction

Circular RNAs (CircRNAs) are single-stranded RNA formed by reverse splicing of precursor mRNA (pre-RNA), which are ubiquitous in eukaryotes and play an important role in regulating transcription, RNA alternative splicing, microRNA (miRNA) sponging, protein translation, and the expression of their parent genes [1,2,3]. CircRNAs do not have the 5′-end caps and 3′-end poly (A) tails; thus, they are not easily degraded by RNA enzymes and have higher stability, resulting in increased attention from researchers [3,4,5]. Research on circRNAs in plants has mainly focused on the identification and characterization of circRNAs [5]. In recent years, circRNAs from nearly 30 plant species, including *Gossypium hirsutum* [4], *Arabidopsis thaliana* [6,7,8], *Oryza sativa* [6,9], *Zea mays* [8,10], *Phyllostachys edulis* [11], *Pyrus betulifolia* [12], *Populus* [5,13], and *Eucalyptus grandis* [14], have been identified. In addition, previous studies have shown that circRNAs might play important roles in regulating plant growth and development [13,15,16], wood formation [17,18], and abiotic stress [8,16,19].

Heterosis is the phenomenon in which F1 hybrids are superior to their parents in terms of yield, quality, resistance, and other traits, which is the main basis for hybrid breeding [4,20,21]. Poplar is a model species of woody plants, which has the characteristics of fast growth, easy reproduction, high yield, and wide adaptability, playing an important role in wood processing, industrial paper making, and forest carbon fixation [5,22]. The main method of poplar breeding is to use heterosis for hybrid breeding and then select good genotypes. However, the mechanism of heterosis formation is still unclear [20,21]. To date, many genetic models have been proposed to explain the mechanism of heterosis formation, including the dominance hypothesis, the overdominance hypothesis, and the epistasis hypothesis; however, none of these hypotheses fully explain the mechanism of heterosis formation [23,24]. With the development of transcriptomics, epigenetics, proteomics, and metabonomics, many studies have attempted to explain the formation of heterosis at the molecular level. mRNAs [25], miRNA [26,27,28], long noncoding RNAs (lncRNAs) [29,30], and circRNAs [4,10] have been confirmed to be related to plant heterosis. The difference in gene expression between parents and their offspring in plants can partly explain the formation of heterosis [20,21,31,32]. Non-additive expression and single-parent expression (SPE) are special differential expression types that have been proved to play an important role in regulating of plant heterosis [32,33,34]. To date, mRNAs, miRNAs, and epigenetic modifications have been confirmed to be involved in the formation of heterosis in poplar [24,25,35,36]; however, the role of circRNAs in poplar heterosis has not been reported.

In this study, we selected the leaves of F1 hybrids with high growth potential (Y3H1, Y3H2, and Y3H3), F1 hybrids with low growth potential (Y3L3 and Y3L4), and their parents (*P. deltoides* cv. ‘55/65’ × *P. deltoides* cl. ‘10/17’) of 3-year-old *Populus deltoides* for circRNA sequencing. We analyzed the characteristics and expression patterns of circRNAs of F1 hybrids with different growth potentials and their parents and explored the possible role of non-additively expressed circRNAs, SPE circRNAs, and the significant differential expression of circRNAs in the regulation of poplar heterosis. We identified key circRNAs that regulate poplar heterosis and their functions, especially the key biological processes, and revealed the molecular mechanism of heterosis formation of a woody plant from a new perspective.

## 2. Results

### 2.1. Identification and Characterization of CircRNA in Poplar’s Leaves

To explore the characteristics of circRNAs in the leaves of poplar F1 hybrids with different growth potentials and their parents, we sequenced the circRNAs from the leaves of the F1 hybrids and their parents of 3-year-old poplar and obtained 21 circRNA expression profiles. The sequencing data comprised 106.09–108.84 Mb of clean reads and 10.61–10.88 Gb of clean bases in each sample. More than 98.00% of the raw reads were retained for subsequent analysis (Appendix A). We identified 3722 circRNAs, of which 908–1051 were expressed in each genotype, and 155 were co-expressed, indicating that most of the circRNAs were genotype-specific (Appendix A, Figure 1A). According to the position of the circRNAs in the genome, they can be divided into four types: intergenic, exonic, intronic, and exon_intron. Among them, the number of circRNAs (57.2%) located in intergenic regions was the largest, followed by exonic (40.2%), exon_intron (1.6%), and intronic (1.0%) (Figure 1B). According to the expression level of circRNAs, they could be divided into three categories: low expression (<100 back scattered reads per million mapped reads (RPM)), moderate expression (100–500 RPM), and high expression (>500 RPM) [37]. The newly identified circRNAs mainly showed a low expression level, and there were fewer circRNAs with high expression level, while the co-expressed circRNAs of the various genotypes were mainly expressed at a high level (Figure 1C,D). In addition, the expression of circRNAs in the F1 hybrids with low growth potential (Y3L3 and Y3L4) was lower than that in the F1 hybrids with high growth potential (Y3H1, Y3H2, and Y3H3) (Figure 1C,D). All identified circRNAs were compared to 1594 parent genes, among which most parent genes (1081, 83.96%) showed a one-to-one correspondence circRNAs, followed by two circRNAs corresponding to one parent gene (156, 12.11%) and more than two circRNAs corresponding to one parent gene (2–24, 0.16–1.86%) (Figure 1E). In addition, the circRNAs were mainly 100–700 bp in length (Figure 1F). 

### 2.2. Differential Expression of CircRNAs in Parents and F1 Hybrids with Different Growth Potential

#### 2.2.1. Expression Differences for Additively Expressed CircRNAs and Non-Additively Expressed CircRNAs between Parents and F1 Hybrids

The difference in the expression levels of circRNAs between the F1 hybrids and their parents is believed to be an important reason for the formation of heterosis, which is worthy of in-depth study [4,10]. Non-additive expression of circRNAs is represented by differentially expressed circRNAs (DECs) between the F1 hybrids and the middle parent value (MPV), which has been proven to be an important reason for the difference between the F1 hybrids and their parents [20]. In this study, compared with the MPV, most of the expression levels of circRNAs (1972–2121) in the F1 hybrids had no significant difference, showing an additive effect. Only a small number of circRNAs (98–237) had significant differential expression, showing a non-additive effect (Figure 2A), indicating that circRNAs of the F1 hybrids mainly show additive genetic effects. Luo et al. found that linear and circular transcripts were mainly non-additive in the leaves of F1 hybrids and their parents of maize [10]. The proportion of additive expression of circRNAs in the F1 hybrids with high growth potential was 91.38–95.44%, and that of non-additively expressed circRNAs was 4.56–8.62%. The percentage of additively expressed circRNAs in the F1 hybrids with low growth potential was 89.84–90.82%, and the percentage of non-additively expressed circRNAs was 9.18–10.16%, indicating that the proportion of non-additively expressed circRNAs in the F1 hybrids with high growth potential was smaller than that in the F1 hybrids with low growth potential (Figure 2B).

Non-additive expression of circRNAs affects the formation of plant heterosis [34]. According to the difference expression levels of circRNAs between parents and the F1 hybrids, we further divided the non-additively expressed circRNAs into four categories: above-high parent expression, high parent expression, low parent expression, and below-low parent expression. In the non-additively expressed circRNAs of the F1 hybrids, the number of circRNAs whose expression level was lower (the sum of the number of circRNAs of low parent expression and below-low parent expression; 55–173) than that of the MPV was higher, while the number of circRNAs whose expression level was higher (the sum of the number of circRNAs of above-high parent expression and high parent expression; 43–94) than that of the MPV was lower (Appendix A). In addition, the expression of circRNAs in the F1 hybrids was significantly higher than the MPV, which was mainly the above-high parent expression (accounting for 26.58–47.00% of the non-additively expressed circRNAs in the F1 hybrids), and the proportion of circRNAs in the high parent expression (accounting for 0.00–1.59% of the non-additively expressed circRNAs in F1 hybrids) was small (Appendix A). Among the F1 hybrids with high growth potential, there were 99 non-additively expressed circRNAs (common to at least two F1 hybrids), compared with five parent genes, which are involved in RNA binding (gene ontology (GO): 0003723), translation (GO: 0006412), glucose metabolism (GO: 0004615, GO: 0006013, and GO: 0009298), and signal transformation (GO: 007165) (Figure 2C; Appendix A). There were 129 non-additively expressed circRNAs (common to at least two F1 hybrids with low growth potential), compared with 18 parent genes (Figure 2C; Appendix A), which are mainly involved in photosynthesis (GO: 0009523, GO: 0015979, and GO: 0009654) and responses to the environment, bacteria, and fungi (GO: 0009266, GO: 0009414, GO: 0071446, and GO: 0009817). These results indicated that there were differences in parent genes with non-additive expression of circRNAs between the F1 hybrids with high growth potential and the F1 hybrids with low growth potential.

#### 2.2.2. Differential Expression of DECs in the Parents and the F1 Hybrids Groups

To reveal the possible role of DECs in the heterosis of the F1 hybrids with different growth potentials and their parents, we assessed the expression differences of circRNAs between the F1 hybrids and their parents. We found that there were 106–156 DECs in the control group of F1 hybrids and their male parents. Among them, 49–106 DECs were upregulated, and 50–72 DECs were downregulated (Figure 2D). In the comparison between the F1 hybrids and female parents, 68–190 circRNAs were significantly differentially expressed. Among them, 33–104 DECs were upregulated, and 35–126 DECs were downregulated (Figure 2D). Except in Y3H3, the number of DECs in the F1 hybrids with low growth potential compared with the female parent group (161–187) was higher than that in the F1 hybrids with high growth potential compared with the female parent group (147–156), while the number of DECs in the F1 hybrids with low growth potential compared with the male parent group (68–97) was lower than that in the F1 hybrids with high growth potential compared with the male parent group (106–128) (Figure 2D). The inconsistency between Y3H3 and other F1 hybrids with high growth potential might be related to the partial male parent of circRNA expression. In the control group of the high growth potential F1 hybrids and the male parent, the numbers of DECs is 97 (common to at least two control groups). In the control group of the high growth potential F1 hybrids and the female parent, the numbers of DECs is 76 (common to at least two control groups) (Figure 2E). The numbers of common DECs between the F1 hybrids with low growth potential and the male parent and between the F1 hybrids with low growth potential and the female parent were 77 and 102, respectively (Figure 2F). These results showed that the expression level of circRNAs in the F1 hybrids with high growth potential was higher than that in the F1 hybrids with low growth potential. We conducted GO function enrichment analysis on parent genes of the DECs of the F1 hybrids with high growth potential and the parents and the F1 hybrids with low growth potential and the parents, respectively (*p* value ≤ 0.05). The enrichment results of the parent genes of DECs in the control group between the F1 hybrids with high growth potential and their parents and between the F1 hybrids with low growth potential and their parents were very similar and were mainly concentrated in transcription, sugar metabolism, amino acid metabolism, energy metabolism, photosynthesis, and responses to abiotic stimuli (Appendix A). These biological processes are closely related to plant growth and development, indicating that DECs between the F1 hybrids with different growth potentials and their parents might participate in the formation of poplar heterosis.

### 2.3. Different Expression of SPE, CoPE and SFE CircRNAs between Parents and the F1 Hybrids

#### 2.3.1. Expression Differences of SPE CircRNAs in the F1 Hybrids and Their Parents

SPE circRNAs are a special type of differentially expressed circRNAs between parents, which have been proven to play an important role in the formation of plant heterosis [34]. We identified 1141 SPE circRNAs from the leaves of parents, including 612 SPE-F (female) circRNAs and 529 SPE-M (male) circRNAs (Appendix A, Figure 3A). The number of SPE circRNAs in the parents (1141) is higher than that of the F1 hybrids (168–230). In addition, SPE-F circRNAs and SPE-M circRNAs were expressed in the F1 hybrids (Figure 3A): 76–96 SPE-M circRNAs and 105–135 SPE-F circRNAs were expressed in the F1 hybrids with high growth, and 89 SPE-M circRNAs and 79–99 SPE-F circRNAs were expressed in the F1 hybrids with low growth (Figure 3A). These results indicated that the expression of SPE circRNAs in the F1 hybrids with different growth potentials has a certain bias. The offspring with high growth potentials show a preference for the female parent’s SPE circRNAs, and the SPE circRNAs of the female parent with high growth potentials might positively regulate the formation of poplar heterosis.

Allelic differences between the parents are the main reason for heterosis, and the differential expression of genes between parents is an important manifestation of allelic differences [4,27]. To further explore the role of SPE circRNAs with significant differential expression between parents in heterosis, we identified 119 DECs among parents using the female parent as the control. Among them, 46 DECs were upregulated and 63 DECs were downregulated (Appendix A). We statistically analyzed the number and expression level of SPE circRNAs in the F1 hybrids with significant difference between the male parent and female parent control groups and found that there were 92 SPE circRNAs in 119 DECs, among which the number of SPE-F circRNAs (60) was more than that of SPE-M circRNAs (32) (Figure 3B). In addition to Y3H3, the number of SPE circRNAs in F1 hybrids with high growth potential (44–51) was more than that in the F1 hybrids with low growth potential (27–33) and in the male parents (32), while the number of SPE-M circRNAs in the F1 hybrids with low growth potential (14–15) was more than that in the F1 hybrids with high growth potential (8) (Figure 3B). Except for Y3H3, the expression pattern of SPE-M circRNAs of the F1 hybrids with high growth potential was similar to that of the female parent, while that of the F1 hybrids with low growth potential was similar to that of male parent (Figure 3C). These results further confirmed that SPE circRNAs might participate in the formation of poplar heterosis. SPE-DECs (male parent and female parent comparison group) identified parent genes, most of which were annotated as resistance proteins or enzymes, such as TMV resistance protein N isoform X1, DNA directed RNA polymerase III subunit 2 isoform X1, ubiquitin old modifier 1 isoform X1, and UDP N-acetylglucosamine transfer subunit ALG14 homolog isoform X1 (Appendix A). The functions of these parent genes were annotated in the main biological processes as being related to plant growth and development, including transcription (GO: 0003677, GO: 0005666, and GO: 0032549), signal transduction (GO: 007165), programmed cell death (GO: 0043068), metal ion binding (GO: 0046872), and photocooperation (GO: 0048564) (Appendix A). In the DECs of the male parent and female parent contrast group, 11 SPE circRNAs were shared by the F1 hybrids with high growth potential and 15 SPE circRNAs were shared by the F1 hybrids with low growth potential. The expression patterns of these SPE circRNAs in the F1 hybrids with the same growth potential were similar (Figure 3D,E). The above results indicated that the SPE circRNAs in the DECs between parents might participate in the formation of poplar heterosis.

#### 2.3.2. Expression Differences of Co-Expressed CircRNAs in the Parents and the F1 Hybrids

To explore whether co-expressed circRNAs in the parents (CoPE) participate in the formation of poplar heterosis, we analyzed their expression in the F1 hybrids. We identified 392 co-expressed circRNAs in parents, and most of them (258–278) were expressed in the F1 hybrids, of which 155 were co-expressed in the F1 hybrids (Figure 4A), indicating that the co-expressed circRNAs among parents were relatively conserved in the F1 hybrids. Co-expressed circRNAs in the parents were mainly highly expressed in the parents and F1 hybrids, and the expression level of the circRNAs in the F1 hybrids with high growth potential was higher than that in the F1 hybrids with low growth potential (Figure 4B). Among the co-expressed circRNAs in the parents, the expression patterns of the male parent and female parent were highly consistent, and only 27 circRNAs were significantly differentially expressed between the parents, mainly with significantly upregulated expression (24) (Appendix A, Appendix A). In addition, there was no significant difference in the expression pattern of co-expressed circRNAs of the parents in the F1 hybrids with different growth potentials (Appendix A). These results indicated that the co-expressed circRNAs between the parents might not participate in the formation of poplar heterosis.

#### 2.3.3. Expression Differences of F1 Hybrid-Specifically Expressed of CircRNAs in F1 Hybrids

SFE circRNAs are specifically expressed in the F1 hybrids, which are not expressed by parents. To explore whether the SFE circRNAs are involved in the formation of poplar heterosis, we analyzed their expression patterns in the F1 hybrids. In this study, most of the identified F1 hybrid-specific circRNAs (2189, 58.81%) had a low expression level (Appendix A). The number of circRNAs in each genotype of the F1 hybrids was 439–587 (Figure 4C), and the expression levels of circRNAs in the F1 hybrids with high growth potential were higher than those in the F1 hybrids with low growth potential (Figure 4D). The expression patterns of F1 hybrid-specific circRNAs were mainly genotype specific (315–413, 67.97–71.95%). Among F1 hybrids with the same growth potential, the co-expression of F1 hybrid specific circRNAs was less. In F1 hybrids with different growth potential, the co-expression of F1 hybrid specific circRNAs is also less (Figure 4C). In addition, the expression patterns of the F1 hybrid-specific circRNAs in the F1 hybrids with the same growth potential were inconsistent (Appendix A). These results suggested that the F1 hybrid-specific circRNAs may not be involved in the formation of poplar’s heterosis.

### 2.4. Expression Differences of CircRNAs among the F1 Hybrids with Different Growth Potentials

To explore the possible role of differentially expressed circRNAs in the F1 hybrids with different growth potentials in poplar growth, we selected F1 hybrids with high growth potential (Y3H1 and Y3H2) and low growth potential (Y3L3 and Y3L4) with consistent expression patterns for differential expression analysis. The analysis results showed that there were 162–236 DECs in Y3L3 vs. Y3H1, Y3L3 vs. Y3H2, Y3L4 vs. Y3H1, and Y3L4 vs. Y3H2, and from which there were 61 were co-DECs (Figure 5A). Among these 61 co-DECs, 34 were all upregulated, and 25 were all downregulated, and the other two were upregulated or downregulated in the four comparative groups (Appendix A). In addition, the expression patterns of the circRNAs in F1 hybrids with high growth potential and female parents were highly consistent, while those of the F1 hybrids with low growth potential and male parents were highly consistent (Figure 5B). In the four comparison groups, DECs were compared with 14–31 parent genes (Figure 5C), and the functional annotation results of these genes showed high consistency, mainly referring to processes related to plant growth and development, including transcription, folding, sorting and degradation, carbohydrate metabolism, energy metabolism, and environmental adaptation. Moreover, there were eight parent genes of common DECs in the four comparison groups, which are mainly involved in the response to stimulus, metadata process, transporter activity, and catalytic activity, such as Podel.08G216500, Podel.10G161200, Podel.13G121900, and Podel.08G176400 (Appendix A, Appendix A). Forty-two common DECs were expressed in the parents, including 21 SPE circRNAs, mainly SPE-F circRNAs (17) (Figure 5D). This indicated that the circRNAs expressed in the parents, especially the SPE circRNAs of the female parent with high growth potential, among the DECs of the F1 hybrids with different growth potentials, might play an important role in regulating of poplar growth potential.

### 2.5. CircRNAs Regulate the Formation of Different Growth Potentials of Poplars by Acting as miRNA Sponges

To explore whether circRNAs regulate the formation of poplars with different growth potentials through miRNA sponging, we predicted the miRNA targeted binding relationship of the identified circRNAs. We identified that 2059 circRNAs and 295 miRNAs that have targeted binding relationships (Appendix A). Among them, 544 circRNAs have only one miRNA binding site, while 1515 circRNAs have multiple miRNA binding sites (2–184) (Figure 6A). A single circRNA can target multiple miRNAs, and a single miRNA can also be targeted by different circRNAs. For example, ath-miR5021 can be targeted by 1052 circRNAs, and ath-miR156h can be targeted by 736 circRNAs (Appendix A). To further explore whether the circRNAs that might be related to poplar heterosis act as miRNA sponges, we conducted statistical analysis on the number of miRNAs targeted. We found that common non-additively expressed circRNAs of the F1 hybrids with high growth potential, common non-additively expressed circRNA of the F1 hybrids with low growth potential, SPE circRNAs, and common DECs of the F1 hybrid with different growth potentials targeted most miRNAs (92.2–100.0% of miRNAs targeted by all identified circRNAs) (Figure 6B). The common non-additively expressed circRNAs of the F1 hybrids with high growth potential, non-additively expressed circRNAs of the F1 hybrids with low growth potential, and common DECs of F1 hybrids with different growth potentials could target 93.0–98.0% of the identified miRNAs, which was much higher than the proportion of all identified circRNA (55.3%) and SPE circRNA (56.9–57.5%) (Figure 6B). Among the different types of miRNA, more members of the miR156, miR157, and miR169 families were targeted (Appendix A). Some members of these families have been confirmed to affect the plant stress response, organ development, growth phase transformation, and phenological regulation by regulating the expression of their target genes [38,39,40]. miR5021, miR156, miR5641, miR5649, and miR5998 were all targeted by a large number of circRNAs (Appendix A). These miRNAs and their corresponding circRNAs might be involved in the formation of poplar heterosis. The above results indicated that non-additively expressed circRNAs might participate in the formation of poplar heterosis through miRNA sponging, and common DECs of the F1 hybrid with different growth potentials might participate in the formation of different growth potentials of F1 hybrids through miRNA sponging. 

### 2.6. Prediction of the Protein Coding Potential of CircRNA in Poplar Leaves

An internal ribosome entry site (IRES) and an open reading frame (ORF) are important functional elements of circRNAs for translation into proteins. Therefore, to predict whether circRNAs could encode proteins, it was necessary to predict whether they contain an IRES and an ORF. The results showed that the vast majority of circRNAs (3001) have an IRES, and 515 circRNAs have one or more ORFs. Among them, 442 circRNAs have both an IRES and an ORF and thus might have protein coding potential (Appendix A). The sequence length of the ORFs in the circRNAs was 63–2298 bp, mainly 63–500 bp (Figure 6C). To further explore whether the newly discovered circRNAs might be related to heterosis function through protein coding, we made a statistical analysis of the number and proportion of circRNAs with protein coding potential. We found that the proportion of common non-additively expressed circRNAs of the F1 hybrids with high growth potential (3.0%), common non-additively expressed circRNAs of F1 hybrids with low growth potential (0.0%), and common DECs of the F1 hybrids with different growth potentials (4.9%) with protein coding potential was lower than among all identified circRNAs (11.3%) (Figure 6D), indicating that these types of circRNAs might not function to regulate poplar heterosis by encoding proteins. The proportion of SPE-M circRNAs and SPE-F circRNAs with protein coding potential was different. The proportion of SPE-F circRNAs with protein coding potential (9.2%) was lower than that of all identified circRNAs, while the protein coding potential of SPE-M circRNAs (12.7%) was higher than that of all identified circRNAs (Figure 6D), indicating that SPE circRNAs might regulate the formation of poplar heterosis by encoding proteins.

### 2.7. Correlation between CircRNA Expression and Their Parental Gene Expression

CircRNAs can affect plant growth and development by regulating the expression of their parent genes [41]. To explore whether the newly discovered circRNAs that may be related to heterosis participate in the formation of poplar heterosis by regulating the expression of their parent genes, we calculated the paired expression correlation between circRNAs and their parent genes in all samples and then counted the proportion of paired expression correlation between each type of circRNA and their parent genes. The positive correlation ratio (47.31%) between all identified circRNAs and their parent gene expression was slightly lower than the negative correlation ratio (52.69%), while the significant positive correlation ratio (72.97%) was much higher than the significant negative correlation ratio (27.03%) (Appendix A). In addition, the positive correlation ratios (50.00–66.67%) between the common non-additively expressed circRNAs of the F1 hybrids with high growth potential, common non-additively expressed circRNAs of the F1 hybrids with low growth potential, SPE circRNAs, and common DECs of the F1 hybrid with different growth potentials were higher than or equal to the negative correlation ratios (33.33–50.00%) (Appendix A). These results indicated that circRNAs mainly regulate the expression of their parent genes positively. In contrast to all circRNAs, SPE-F circRNAs and their parent genes were mainly positively correlated (59.88%), while SPE-M circRNAs and their parent genes are mainly negatively correlated (61.20%) (Figure 7A), indicating that SPE-F circRNAs might positively regulate the expression of their parent genes, while SPE-M circRNAs might negatively regulate the expression of their parent genes. Except for the common non-additively expressed circRNAs of the F1 hybrids with high growth potential, the proportion of other types of circRNAs significantly related to their parent genes (8.14–16.67%) was higher than that of all identified circRNAs (5.77%) (Appendix A), indicating that circRNAs might regulate poplar heterosis by regulating the expression of their parent genes. The proportion of positively correlated SPE-F circRNAs was 92.86%, and the proportion of negatively correlated SPE-F circRNAs was 7.14%; The proportion of positively correlated SPE-M circRNAs was 12.50%, and the proportion of negatively correlated was SPE-M circRNAs 87.50% (Appendix A). This indicated that SPE-F circRNAs might regulate poplar heterosis by positively regulating the expression of their parent genes, while SPE-M circRNAs might regulate poplar heterosis by negatively regulating the expression of their parent genes. In addition, we conducted functional enrichment analysis on the parent genes significantly related to circRNA expression (Appendix A) and found that the significantly positively related parent genes were mainly involved in carbohydrate metabolism, photosynthesis, and energy conversion, while the significantly negatively related parent genes were mainly involved in transcription, translation, and material transport. This suggested that circRNAs might affect various biological processes (photosynthesis pathway and energy metabolism) of plants by regulating the expression of their parent genes and then regulate the growth and development of poplar to form different growth potentials. 

### 2.8. Co-Expression of Protein Coding Genes and CircRNAs

To explore the role of circRNAs in poplar, especially in regulating heterosis, we first combined the fragments per kilobase of exon per million mapped fragments (FPKM) matrix of significantly differentially expressed genes (DEGs) and the RPM matrix of DECs of 11 comparison groups between the parents and F1 hybrids and between the parents to conduct weighted correlation network analysis (WGCNA) to identify the modules related to different growth potentials of poplar and then explore the key biological processes and key circRNA regulating poplar heterosis. We selected significant DEGs (17046) and DECs (417) (At least one control group of the 11 control groups between parents and F1 hybrids and between male parent and female parent) for WGCNA analysis and obtained 24 modules, with an average of 727.6 DEGs or DECs per module (Figure 7B). After calculating the correlation between each module and genotype, we found that the correlation between the F1 hybrids with the same growth potential and each module was more similar, compared with F1 hybrids with different growth potential (Figure 7B). Such as, the pink module was significant positively correlated (0.52) in the high growth potential offspring (Y3H1 and Y3H2) and positively correlated (0.34) in the female parent, while negatively correlated (−0.27 to −0.39) in the male parent and low growth potential offspring (Y3L3 and Y3L4), which indicated that DEGs and DECs in the pink module might positively regulate the high growth potential of poplar (Figure 7B). Otherwise, the dark red module was significantly positively correlated (0.55 to 0.70) in low growth potential offspring but negatively correlated (−0.22 to −0.30) in the other genotypes, suggesting that DEGs and DECs in the dark red module might negatively regulate the growth potential of poplar (Figure 7B).

To explore the role of pink and dark red modules in regulating poplar heterosis, we conducted GO and Kyoto Encyclopedia of Genes and Genomes (KEGG) functional enrichment analysis (*p* value ≤ 0.05) on the DEGs of these two modules. We found that the pink module was mainly enriched in carbohydrate metabolism, amino acid metabolism, material transport, and epigenetic modification (Appendix A), while the dark red module was mainly enriched in amino acid metabolism, carbohydrate metabolism, energy metabolism, plant hormone signal transduction, and plant pathogen interaction (Appendix A), indicating that regulation of these biological processes might regulate the formation of different growth potentials of F1 hybrid poplars. Based on the co-expression relationship between DEGs and DECs, we inferred that circRNAs in the pink and dark red modules might play an important role in regulating poplar heterosis. There are 25 DECs co-expressed with DEGs in the pink module and 12 DECs co-expressed with DEGs in the dark red module (Appendix A). These DECs might affect the heterosis of poplar by regulating the biological processes closely related to plant growth and development, such as carbohydrate metabolism and amino acid metabolism. Except for the common non-additively expressed circRNAs of the F1 hybrids with high growth potential, which had only one representative in the dark red module (1), the common non-additively expressed circRNAs of the F1 hybrids with high growth potential (11, in pink module), common non-additive circRNAs of low growth potential F1 hybrids (11–14), and common DECs of F1 hybrid with different growth potentials (7–17) had more representatives in both modules (Figure 7C). There were significantly more SPE circRNAs in the pink module (17) than in dark red module (1), especially SPE-F circRNAs. This analysis confirmed that SPE-F circRNAs might positively regulate the high growth potential of poplar (Figure 7C). Based on the correlation of gene expression, we used the Cytoscape software to visualize the co-expression network of the pink module and the dark red module and screened 25 hub DEGs or hub DECs, respectively. At last, two hub DECs (novel_circ_000368 and novel_circ_001540) were identified in pink module, and five hub DECs (novel_circ_001330, novel_circ_003614, novel_circ_001672, novel_circ_002310, and novel_circ_003239) were identified in dark red module (Appendix A; Figure 7D, E). According to the above expression pattern analysis, these seven circRNAs are non-additively expressed circRNAs (Appendix A), which further indicates that non-additively expressed circRNAs might play an important role in the formation of different growth potentials of poplar. On the basis of GO and KEGG functional enrichment analysis of pink module and dark red module, we speculate that these seven hub circRNAs may regulate poplar heterosis through carbohydrate metabolism, amino acid metabolism, energy metabolism, and material transport. However, their specific functions in poplar heterosis require further analysis and verification.

## 3. Discussion

### 3.1. Expression Level and Source of CircRNAs in Leaves of F1 Poplar Hybrids with Different Growth Potentials and Their Parents

Heterosis is the main theoretical basis for hybrid breeding of plants, and the excellent varieties of forests in the world are mainly selected and cultivated through artificial hybridization [4,20]. Although much research has been carried out on the formation mechanism of plant heterosis [2,4,26,27], the molecular explanation of its formation mechanism is unclear [23,24]. In this study, we used poplar, the model species of woody plants, as the material to analyze the differences between offspring with different growth potentials and their parents from the perspective of circRNAs and explored the mechanism of heterosis formation in this forest tree. We identified 3722 circRNAs from the leaves of the F1 hybrids and their parents, and the number of circRNAs in each genotype was different. The sequence length of the identified circRNAs is mainly 100~700 bp, and most of the circRNAs were expressed at a low level, which is consistent with previous research of *Gossypium hirsutum* and poplar [4,5]. The expression of circRNAs in the F1 hybrids with high growth potential was higher than that in the F1 hybrids with low growth potential. The co-expressed circRNAs in all genotypes were mainly expressed at high levels, and their expression in the F1 hybrids with high growth potential was also higher than that in the F1 hybrids with low growth potential, indicating that the high expression of circRNAs could be related to the high growth potential of the F1 hybrids. We also found that the circRNAs of leaves in poplar were mainly derived from to the intergenic and exon regions, with fewer being derived from introns, which was consistent with the research results of Liu et al. [42]. Zhou et al. [5] found that circRNAs of poplar roots treated with different forms of nitrogen were mainly mapped to exons; Li et al. [19] also found that circRNAs in leaves, phloem, xylem, and roots of poplar were mainly mapped to exons. In addition, the circRNAs of *Oryza sativa* [9], *Arabidopsis thaliana* [43], and *Polyploid Gossypium* [4,44] were mainly mapped to exons, while in *Glycine max* [45] they mainly mapped to exon_intron. In *Solanum lycopersicum* [46] and *Actinidia* [41] they were mainly intergenic. At present, there is no uniform rule for the source distribution of circRNAs, which might be affected by species, tissue parts, treatment methods, and identification methods [4,18,47]. Therefore, using leaves of poplar as materials, we found that the formation of different growth potentials of F1 hybrids might be related to intergenic and exonic circRNAs; however, more studies on tissue parts are needed to clarify the circRNAs that play important roles in poplar heterosis and to provide valuable information for subsequent molecular breeding.

### 3.2. Differential Expression of CircRNAs in Leaves of F1 Hybrid Poplars with Different Growth Potentials and Their Parents

Genes that control a trait tend to have similar expression patterns in plant individuals whose external traits are similar [48]. The non-additively expressed circRNAs has been confirmed to be widely involved in the formation of plant heterosis [4]. In this study, the proportion of non-additively expressed circRNAs in the F1 hybrids was relatively small, while the proportion of additively expressed circRNAs was relatively large, which is consistent with previous research [4,20,27]. In addition, the proportion of non-additively expressed circRNAs in the high growth potential F1 hybrids was lower than that in the low growth potential F1 hybrids, indicating that there were differences in non-additively expressed circRNAs in different growth potential F1 hybrids. The parent genes of these non-additively expressed circRNAs are mainly involved in biological processes closely related to plant heterosis, including carbohydrate metabolism, signal transduction, plant hormone synthesis, and response to stress. We speculated that the expression pattern of circRNAs in the parents might affect the formation of poplar heterosis. We divided the identified circRNAs into three categories according to their parental expression pattern: SPE circRNAs, co-expressed circRNAs in the parents, and circRNA not expressed in the parents. The dominant model of heterosis is realized by complementing beneficial alleles and harmful alleles among the parents [49,50]. SPE is a special form of gene expression complementation between parents, which has been proven to be an important reason for the excellent performance of F1 hybrid traits [4,50]. In this study, the number of SPE circRNAs in the parents was much higher than that in their F1 hybrids, which is inconsistent with previous studies [10,34]. However, the number of SPE circRNAs of the F1 hybrids with high growth potential was higher than that of the F1 hybrids with low growth potential, especially for SPE-F circRNAs. This result indicated that SPE circRNAs, especially SPE-F circRNAs of the female parent with high growth potential, might positively regulate the heterosis of poplar biomass. Baldauf et al. reported similar results [50]. They analyzed the gene co-expression network of six maize inbred lines and found that the modules that correlated positively with phenotypic traits were rich in the SPE-B Type and are mainly involved in plant growth and development [50]. We found that the expression pattern of co-expressed circRNAs in parents and their F1 hybrids was independent of their growth potential, and the expression of most of circRNAs was conserved in the parents and their F1 hybrids. In addition, most of the circRNAs we identified as being expressed specifically in the F1 hybrids had expression patterns that were independent of their growth potential. These results indicated that the co-expressed circRNAs in the parents and the F1 hybrid-specific circRNAs might not participate in the formation of poplar heterosis.

The difference in circRNA expression between the parents and the F1 hybrids is an important reason for the formation of plant heterosis [4,10]. After analyzing the differential expression of circRNAs between the F1 hybrids and their parents, we found that compared with the F1 hybrids with low growth potential, the F1 hybrids with high growth potential and female parent group had fewer DECs and common DECs, which was consistent with the small difference between female parent and F1 hybrids with high growth potential, indicating that circRNAs might be involved in the heterosis of poplar growth. Zhao et al. [51] found that the number of mRNAs with significantly different expression between the F1 hybrids with a high heterosis rate and their parents was higher than that of the F1 hybrids with low heterosis rate, which confirmed that the difference in gene expression between F1 hybrids and their parents was related to heterosis. The differential expression of alleles between parents is an important reason for the differential expression of F1 hybrid traits [4]. We found that the expression patterns of DECs between parents in the F1 hybrids and parents with similar growth potentials were highly consistent. In addition, most of DECs between parents are SPE circRNAs, mainly SPE-F circRNAs, especially in the F1 hybrids with high growth potential, while the number of SPE-M circRNAs in the F1 hybrids with low growth potential was higher than that in the F1 hybrids with high growth potential. These results further indicated that SPE-F circRNAs might positively regulate the heterosis of poplar biomass, while SPE-M circRNAs might negatively regulate the heterosis of poplar biomass. Furthermore, we speculated that the differentially expressed circRNAs in the F1 hybrids with different growth potentials might be involved in regulating the growth potential in poplar. Among the DECs (among four comparison groups of F1 hybrids with high growth potential and F1 hybrids with low growth potential), the expression patterns of the F1 hybrids with high growth potential and female parents and F1 hybrids with low growth potential and male parents were highly consistent, which is also consistent with the performance of growth traits. CircRNAs can affect plant growth by competitively inhibiting the expression of parental genes [1,4]. The DECs of the four comparison groups were compared to eight parent genes, which are mainly involved in biological processes closely related to plant growth and development, such as protein synthesis, stress response, carbohydrate metabolism, signal transduction, and energy conversion. Some examples are the subtilisin-like protein SBT5.3 isoform X2 (Podel.11G045400), TMV resistance protein N isoform X1 (Podel.19G019700), and serine/acute repetitive matrix protein 1 isoform X1 (Podel.03G169100). The subtilisin-like serine protein (*HbSPA*) gene in *Hevea brasiliensis* participates in *Phytophthora Palmivora* infection defense, which is significantly upregulated in infected leaves [52]. The above results indicated that circRNAs, especially SPE circRNAs, might participate in the formation of poplar heterosis, and the DECs between parents might participate in the formation of F1 hybrid growth potential; however, their role in the growth and development of poplar requires further analysis.

### 3.3. Functional Prediction of CircRNAs Related to the Formation of Poplar Heterosis

CircRNAs can affect a variety of biological processes of plants by acting as miRNA sponges, being translated into protein, and by affecting the expression of their parental gene, thus modulating plant growth and development [1,2,3,4]. In this study, we predicted the targeting relationship between circRNAs and miRNAs and the protein coding potential of circRNAs and analyzed the correlation between circRNA expression and their parental gene expression to explore how circRNAs might regulate the growth potential of poplar F1 hybrids. CircRNAs can affect the growth and development of plants by acting as miRNA sponges [1,3]. In this study, we identified a large number of circRNAs and miRNAs that have targeted binding relationships. Among them, the proportions of targeted miRNAs of common non-additively expressed circRNAs in the F1 hybrids with high growth potential, common non-additively expressed circRNAs in the F1 hybrids with low growth potential, and common DECs (from the four comparison groups of F1 hybrids with high growth potential and F1 hybrids with low growth potential) were higher than that of all identified circRNAs, indicating that these types of circRNAs might participate in the formation of different growth potentials of poplar through miRNA sponging. An miRNA that has a target binding relationship with a circRNA might regulate the growth and development of poplar by regulating its response to stress. For example, miR398 and its target partner superoxide dismutase1 (CSD1) are involved in the regulation of abscisic acid (ABA) and salt stress related pathways of poplar and *Arabidopsis thaliana* [53]; PtmiR169o can enhance the drought resistance of poplar by inhibiting the expression of its target gene *PtNF-YA6*, thereby promoting its growth [39]. In addition, we also found that a large number of circRNAs target miRNA family members that play a key role in plant organ development, growth phase transition, and phenological regulation, such as the miR156 and miR157 families [38,40,54,55,56]. miRNA156/157 combines with the miRNA recognition element (MRE) of the *SPL3* gene of *Arabidopsis thaliana* to promote early flowering [38]. The expression of miR156 in poplar decreases with age, and the juvenile stage of transgenic plants overexpressing miR156 was significantly prolonged [40]. It is worth noting that a large number of members of these miRNA families have targeted binding relationships with circRNA. CircRNAs can regulate the growth and development of organisms by being translated into proteins [57,58]. For example, circ-ZNF609 can be translated into a protein that promotes the proliferation of mouse and human muscle cells [57]. CircSEMA4B inhibits the proliferation, migration, and invasion of breast cancer cells by encoding the SEMA4B-211aa protein and can also act as an miR-330-3p sponge to upregulate the expression of the tumor suppressor gene *PDCD4* [58]. There is direct evidence that circRNAs in animals can be translated into proteins; however, there are few reports of their translation in plants [59,60]. Liao et al. [60] found that mitochondrial-encoded circRNAs (mcircRNAs) exist widely in plants, which can bind to ribosomes and be translated into proteins. In this study, we identified a large number of circRNAs with protein coding potential. The proportion of SPE-M circRNAs and SPE-F circRNAs with protein coding potential was high, while the proportion of other types of circRNAs with protein coding potential was low, indicating that SPE circRNAs might participate in the formation of poplar heterosis through protein translation. The identified circRNAs have one or more ORFs, the sequence length of which was mainly 63–500 bp, which is consistent with a previous report [61]. CircRNAs can regulate plant growth and development by regulating the expression of their parent genes [9,46] Our research found that the ratio of the significant positive correlation between a circRNA and its parent gene expression was higher than that of the significant negative correlation, which was consistent with previous research results [41]. The proportion of the types of circRNAs related to heterosis of poplar significantly related to the paired expression of parental genes was higher than that of all identified circRNAs, indicating that these circRNAs might regulate the heterosis of poplar by regulating the expression of their parental genes. In addition, the proportion of significant negatively correlated expression of SPE-M circRNAs-parent gene pairs was higher than that of significant positive correlation, while the proportion of significantly positively correlated expression between other types of circRNA related to heterosis of poplar and their parent genes was higher than that of significant negative correlated pairs, indicating that SPE-M circRNAs can regulate heterosis of poplar by inhibiting the expression of their parent genes, and other types of circRNAs might regulate poplar heterosis by enhancing the expression of their parent genes. CircRNAs in animals can increase the expression of their parent genes via cis-acting elements [1,62]; however, how they enhance or inhibit the expression of the corresponding parent genes in plants to regulate the formation of poplar heterosis requires further research and verification.

To explore the key biological processes and circRNAs that regulate the heterosis of poplar, we used WGCNA to construct a co-expression network of circRNAs and protein coding genes. We identified the pink module and the dark red module, which were significantly positively correlated with the growth of poplar. The protein coding genes of the pink module and the dark red module are mainly involved in carbohydrate metabolism, amino acid metabolism, energy metabolism, material transport, and epigenetic modification, indicating that circRNAs might affect plant heterosis by regulating various biological processes. These results are consistent with those of previous studies [20,63,64]. Interestingly, we found that most of the circRNAs in the pink and dark red modules were screened in the present study and thus might be related to poplar heterosis. Among them, the number of SPE circRNAs in the pink module was large, and most of them are SPE-F circRNAs. This result confirmed that SPE-F circRNAs might positively regulate poplar heterosis. Baldauf et al. [50] also found that the modules in maize that are positively related to the growth traits were rich in genes expressed by the female parent but not by the male parent. We identified novel_circ_001540 and novel_circ_000368, which might positively regulate poplar heterosis in the pink module, and novel_circ_001330, novel_circ_003614, novel_circ_001672, novel_circ_002310, and novel_circ_003239, which might negatively regulate poplar heterosis in the dark red module. These seven circRNAs are non-additively expressed, which further indicated that the non-additively expressed circRNAs might participate in the formation of poplar heterosis; however, their specific functions require further analysis and verification.

## 4. Materials and Methods

### 4.1. Material Source and Field Test Design

We used F1 hybrids with higher tree height growth potential than their parents, F1 hybrids with lower tree height growth potential than their parents, and their parents (the female parent was *P. deltoides* cv. ‘55/65’, the male parent was *P. deltoides* cl. ‘10/17’) genotypes as materials [24,64] to build experimental forests in Jiaozuo City, Henan Province, in March 2019. These F1 hybrids were produced from the same intraspecific cross combination of *Populus deltoids* and verified by field experiments for many years to ensure the stability of the growth potential of each genotype. The experimental site (35°8′ N, 113°17′ E) has a temperate continental climate, with an annual average temperature of 15.2 °C, an extreme maximum temperature of 43.6 °C, and an extreme minimum temperature of 14.3 °C. The average annual precipitation is 625.4 mm; the relative humidity is 61%; the frost-free period is 224 d; the annual lighting hours are 2434 h. The soil is sandy loam with a pH value of 7.2. The trial forest was designed as a completely randomized block with 4 blocks (replicates) and 9 plots. Row and column spacing was 1.8 m × 1.8 m. All poplar genotypes in the experimental forest were under consistent routine management.

For each genotype, three plants with consistent and healthy growth were selected as three biological replicates. For each plant, the 4–6 leaves from the top to the bottom of the current year branches in the southeast and northwest directions at the same height (2/3 of the tree height) were selected for mixed sampling. The samples were stored in liquid nitrogen in the field and in the refrigerator at −80 °C indoors. At the same time, the growth traits (tree height and diameter at breast height (DBH)) of each genotype were measured and analyzed. At the time of sample collection in this study, the average tree height of Y3H1, Y3H2, and Y3H3 was 8.60 m (8.69 m, 8.51 m, and 8.59 m), and the average DBH of them was 6.20 cm (6.55 cm, 5.89 cm, and 6.03 cm), which exhibited heterosis over higher parent for tree height (12.27% for Y3H1, 9.95% for Y3H2, and 10.98% for Y3H3) and DBH (12.93% for Y3H1, 1.55% for Y3H2, and 3.97% for Y3H3). On the other hand, average tree height of Y3L3 and Y3L4 was 7.30m (7.18 m and 7.50m), and the average DBH of them was 4.77 cm (4.63 cm and 4.67 cm), which possessed negative heterosis over higher parent for tree height (−7.24% for L3 and −3.10% for L4) and DBH (−20.17% for Y3L3 and −19.48% for Y3L4). The growth performance of each genotype was consistent with the previous research results [29,62].

### 4.2. CircRNA Sequencing and Identification

We used the cetyltrimethylammonium bromide (CTAB) method to extract total RNA from 21 samples. After digesting the DNA fragments in the total RNA samples with DNase I, further purification and recovery were carried out. The Ribo-off rRNA Depletion Kit (Vazyme Biotech, Nanjing, China) was used to remove rRNA from total RNA. The remaining RNA was fragmented and then used as the template to synthesize the first and second strands of cDNA in turn. Uracil-DNA Glycosylase was used to digest the second strand cDNA, and PCR amplification was then performed. The quality of the constructed small RNA library was tested, and the qualified library products were cyclized and copied to form DNA nanosphere beads (DNBs). Single ended sequencing of the small RNA sequencing library was carried out on the DNBSEQ platform (MGI Tech Co., Ltd., Shenzhen, China), and the average sequencing length was 50 bp.

We used SOAPnuke (v1.5.2) [65] to filter out the reads with contaminated joint, low quality, and an unknown base N content greater than 5% from the raw reads to obtain clean reads. CIRI (v2.0.5) [66] and find circ (v1.2) [67] were used to predict circRNAs, and the results of the two software were integrated according to the start and end positions of circRNA (combing the circRNA whose starting and ending positions are within the first and last 10 bases into one group). These two softwares were used to predict the number of back-spliced reads compared to the two ends of the circRNA (taking the average of outputs of the two software), and finally RPM was used to represent the expression of circRNAs in each sample. According to the position of circRNA on the *Populus deltoides* genome (https://phytozome-next.jgi.doe.gov/info/PdeltoidesWV94_v2_1), we divided circRNAs into four categories: internal, exonic, intronic, and exon_intron. In addition, we used psRobot [68] and TargetFinder [69] to predict the target binding relationship between circRNAs and miRNAs. To explore the protein coding potential of the circRNAs, we used the cORF pipeline (https://github.com/kadenerlab/cORF_pipeline) and IRESfinder [70] (https://github.com/xiaofengsong/IRESfinder) to predict the ORF and IRES of each circRNA, respectively. The analysis of this part is completed by Beijing Genomics institution (BGI; Beijing, China).

### 4.3. Identification of Non-Additively Expressed CircRNAs, SPE CircRNAs, and DECs

To explore the difference expression patterns of circRNAs between the F1 hybrids and their parents, we divided the circRNAs into two types according to the difference expression of circRNAs between F1 hybrids and the MPV ((Y3MP+Y3FP)/2) (Y3MP was expression of circRNAs in male parent; Y3FP was expression of circRNAs in female parent): additive expression and non-additive expression. Among them, circRNAs with significant differential expression between the F1 hybrids and the MPV are considered as non-additively expressed circRNAs, while those with no significant differential expression were additively expressed circRNAs [4,20]. According to the method of Stupar et al. [71], we further divided the non-additively expressed circRNAs into four categories: above high parent expression (AHP), high parent expression (HP), low parent expression (LP), and below low parent expression (BLP). The circRNAs in the F1 hybrids whose expression was significantly higher than the MPV were called AHP circRNAs or HP circRNAs. The expression of AHP circRNAs in the F1 hybrids was higher than that of the parent with high growth potential (female parents). The circRNAs in the F1 hybrids whose expression was significantly lower than the MPV, which were called LP circRNAs or BLP circRNAs. The expression of BLP circRNAs in the F1 hybrids was lower than that of the parent with low growth potential (male parents). SPE circRNAs are expressed in only one parent [4,34]. We divide the SPE circRNAs into two categories: SPE-M circRNAs and SPE-F circRNAs. SPE-M circRNAs are only expressed in the male parent but not in the female parent; SPE-F circRNAs are expressed in the female parent but not in the male parent. In addition, we used DEGseq [72] to analyze the significant differential expression of circRNAs between male parents and female parents, between F1 hybrids and their parents, and between the high growth potential F1 hybrids and the low growth potential F1 hybrids. The difference threshold values were |log2 (Fold Change)| ≥ 1 and q value ≤ 0.05. According to the annotation results of GO and KEGG, we used the phyper function in the R software (https://www.R-project.org/ (BGI; Beijing, China)) to perform functional enrichment analyses on the parent genes of the DECs, and the difference threshold was *p* value ≤ 0.05.

### 4.4. Correlation between the Expression Levels of CircRNAs and Their Parental Genes

To explore whether the circRNAs related to heterosis participate in the formation of heterosis by regulating the expression of their parent’ genes, we used the Pearson method to calculate the paired expression correlation between the circRNAs (RPM) and their parent genes (FPKM) in all samples (*p* ≤ 0.05). We carried out GO and KEGG functional enrichment of parent genes significantly related to circRNA expression through the Omicshare platform (https://www.omicshare.com/tools/ (accessed on 21 November 2022)), and the threshold of significant enrichment was *p* value ≤ 0.05.

### 4.5. Co-Expression of Protein Coding Genes and CircRNAs

We used WGCNA [73] to explore the potential functions of circRNAs involved in poplar heterosis. First, we combined the expression matrix of significant DEGs (17046) and DECs (417) in the parent and F1 hybrid groups and the female parent and male parent groups as the input file of WGCNA to obtain multiple co-expression modules of DEGs and DECs. We identified the key modules that were significantly related to poplar growth (*p* ≤ 0.05) by analyzing the correlation between genotypes and modules. Further GO and KEGG functional enrichment analysis of genes in the key modules was carried out using the Omicshare platform, and the significant enrichment threshold was *p* value ≤ 0.05, from which the function of circRNAs in the corresponding modules was deduced. Finally, we used Cytoscape (v3.7.1) [74] to visualize key modules and filter hub genes and hub circRNAs.

## 5. Conclusions

There were significant differences in circRNA expression among the parents, between the parents and the F1 hybrids with different growth potentials, and between F1 hybrids with different growth potentials. Different types of circRNAs might regulate poplar heterosis in different ways. For example, non-additively expressed circRNAs might regulate poplar heterosis through miRNA sponging; SPE circRNAs might regulate poplar heterosis through protein translation, and DECs among F1 hybrids with different growth potentials might regulate the growth potential of poplars through miRNA sponging. According to the correlation analysis between circRNA expression and its parent gene expression, SPE-M circRNAs might regulate poplar heterosis by inhibiting the expression of their parent genes, while other types of circRNA might regulate poplar heterosis by enhancing the expression of their parent genes. In addition, WGCNA found that circRNAs mainly regulate the heterosis of poplars by participating in carbohydrate metabolism, amino acid metabolism, energy metabolism, material transport, and epigenetic modification. We identified two circRNAs that might be positively regulated and five circRNAs that might be negatively regulated during poplar heterosis.

## Figures and Tables

**Figure 1 ijms-24-02284-f001:**
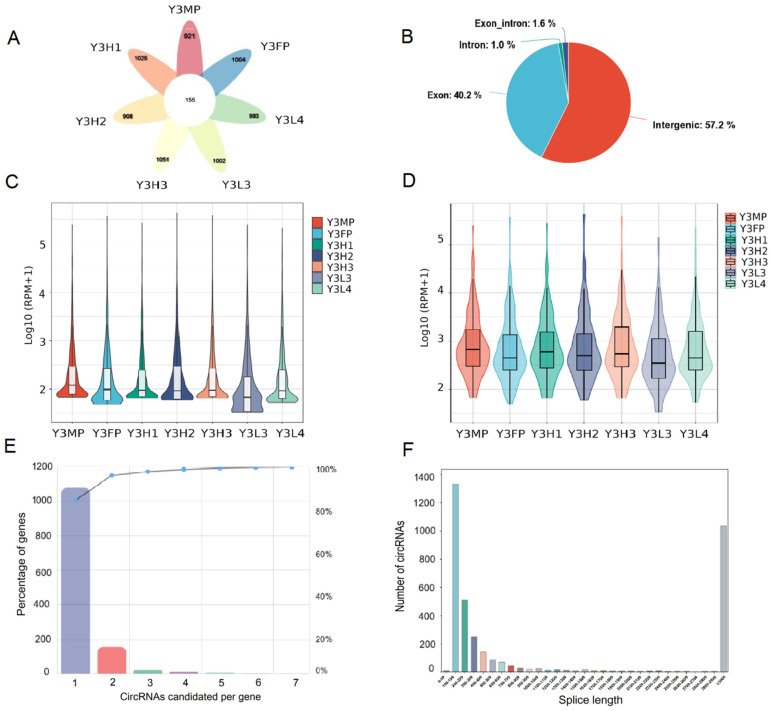
Characteristics of circRNA in the leaves of poplar F1 hybrids and their parents. (**A**) The number of circRNAs in each genotype of poplar. (**B**) Percentage of all identified circRNAs were compared to the genome intergenic regions, exons, exon_intron, and introns. (**C**) The distribution of circRNA expression in each genotype. (**D**) The distribution of the expression levels of circRNAs co-expressed by all genotypes. (**E**) The number of parental genes producing 1–7 circRNAs and the cumulative percentage of parental genes. (**F**) The distribution of sequence length of circRNAs. circRNA, circular RNA; RPM, back scattered reads per million mapped reads; FP, female parent; MP, male parent; H, high growth potential; L, low growth potential.

**Figure 2 ijms-24-02284-f002:**
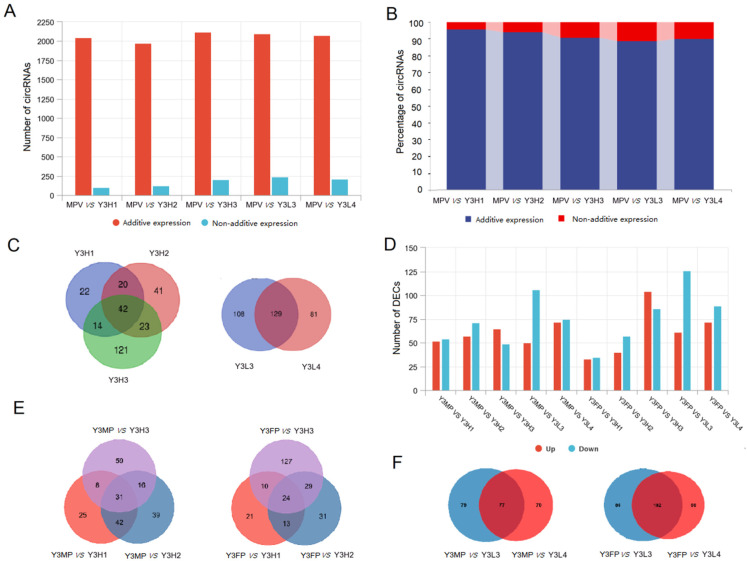
Expression of DECs between parents and F1 hybrids. (**A**) The number of additively expressed circRNAs and non-additively expressed circRNAs in the F1 hybrids and the MPV in contrasting groups. The total number of circRNAs in each control group is the number of circRNAs expressed in F1 hybrids or MPV > 0. (**B**) The ratio of additive expression of circRNAs and non-additive expression of circRNAs in the F1 hybrids and the MPV control group. (**C**) Venn Diagram of non-additively expressed circRNAs in the F1 hybrids with high growth potential and the F1 hybrids with low growth potential (**D**) The number of upregulated and downregulated DECs in the parent and the F1 hybrid control groups. (**E**) Venn Diagram of DECs in the parents and the F1 hybrids with high growth potential control group. (**F**) Venn Diagram of DECs in parents and the F1 hybrids with low growth potential control group. DEC, differentially expressed circRNA; MPV, middle parent value.

**Figure 3 ijms-24-02284-f003:**
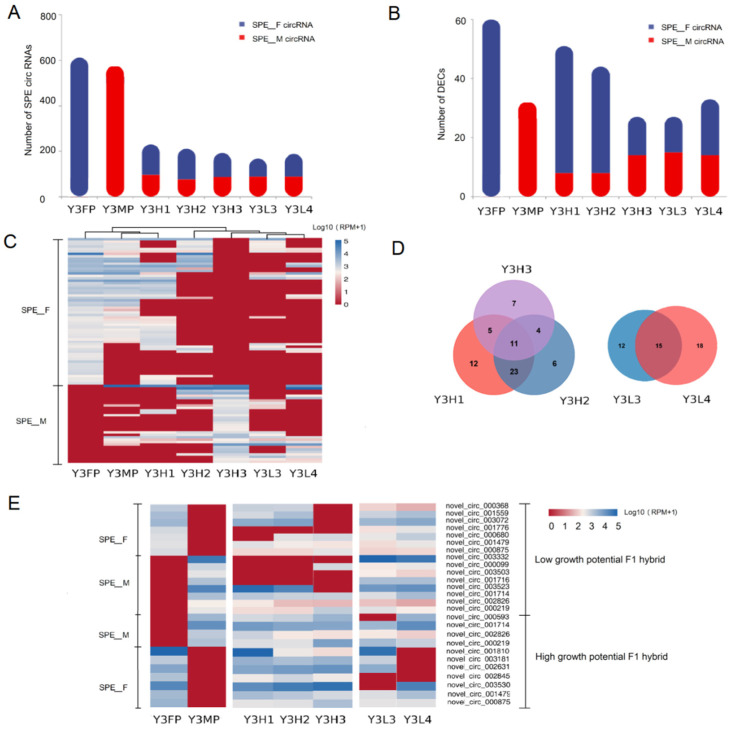
Expression differences of single-parent expression (SPE) circRNAs in the F1 hybrids and their parents. (**A**) The number of SPE-M circRNAs and SPE-F circRNAs in the parents and F1 hybrids. (**B**) The number of SPE-M circRNAs and SPE-F circRNAs among DECs between the parents and the F1 hybrids. (**C**) Heat map of SPE circRNA expression in the F1 hybrids and parents in the DECs between the parents. (**D**) Venn diagram of SPE-DECs in the F1 hybrids with high growth potential and the F1 hybrids with low growth potential. (**E**) Heat map of SPE-DECs shared by the F1 hybrids with high growth potential and the F1 hybrids with low growth potentials.

**Figure 4 ijms-24-02284-f004:**
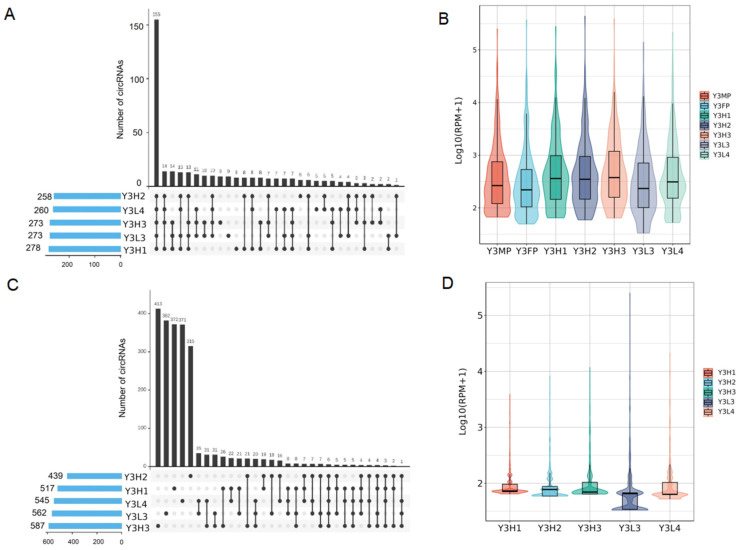
The difference in expression of co-expressed circRNAs of the parents and specific expression of circRNAs of the F1 hybrids in parents and the F1 hybrids (**A**) Among the co-expressed circRNAs of the parents, the number of circRNAs in the parents and F1 hybrids. (**B**) The expression levels of co-expressed circRNAs of the parents in parents and the F1 hybrids. (**C**) The numbers of specifically expressed circRNAs of the F1 hybrids (not expressed by the parents). (**D**) In specific expression of circRNAs of F1 hybrids (not expressed by the parents), the expression level of these circRNAs in F1 hybrids.

**Figure 5 ijms-24-02284-f005:**
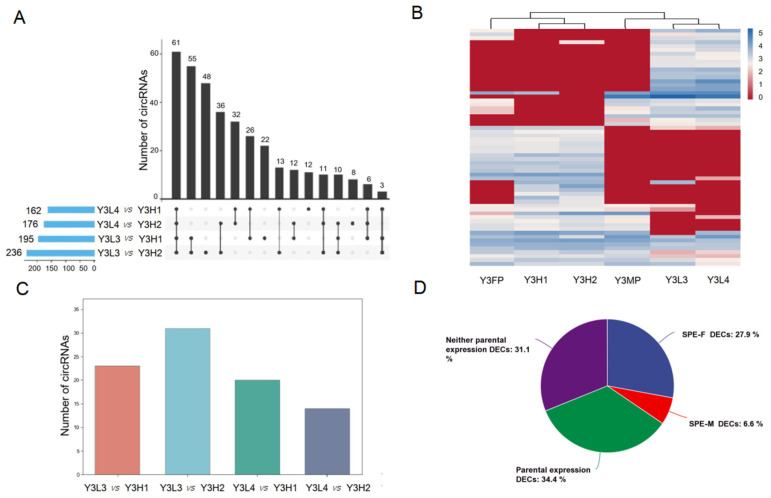
The difference in expression of circRNAs between F1 hybrids. (**A**) The number of DECs in the four comparison groups of F1 hybrids with high growth potential and F1 hybrids with low growth potential. (**B**) Expression of common DECs among the four comparison groups in the parents and F1 hybrids. (**C**) Number of parent genes of the DECs in the comparison groups. (**D**) The proportion of SPEs, expression in neither parent, and parental expression (expressed by both male and female parents) of the DECs in four comparison groups.

**Figure 6 ijms-24-02284-f006:**
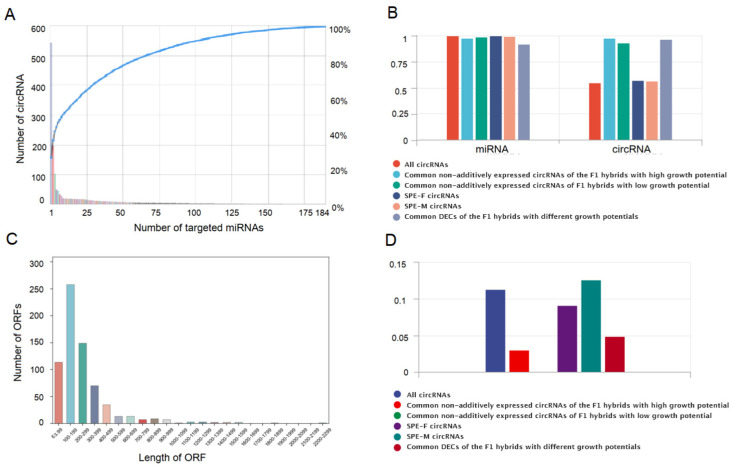
The targeting relationship between circRNAs and microRNAs (miRNAs) and the prediction of protein coding potential of circRNAs. (**A**) The number and distribution of circRNAs that can target 1–184 miRNAs. (**B**) Among different types of circRNAs, the proportion of targeted miRNAs (accounting for the proportion of all targeted miRNAs) and the proportion of circRNAs with targeted relationship with miRNAs. (**C**) The sequence length distribution of open reading frames (ORFs) in the identified circRNAs. (**D**) The proportion of circRNAs with protein coding potential among different types of circRNAs.

**Figure 7 ijms-24-02284-f007:**
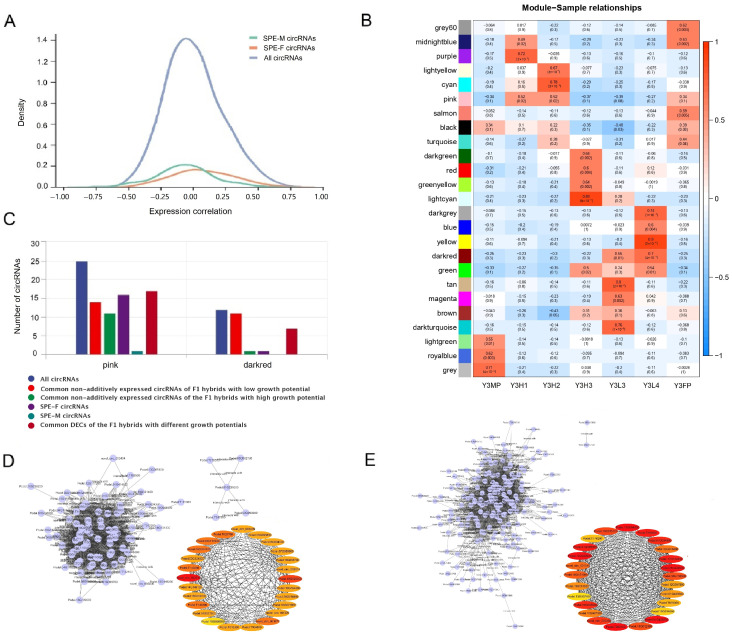
Correlation and weighted correlation network analysis (WGCNA) between circRNA expression and gene expression. (**A**) Density distribution of the correlation coefficients between all circRNA, SPE-M circRNAs, and SPE-F circRNAs and their parent gene expression. (**B**) Correlation between each module and genotype in WGCNA. (**C**) The number of each type of circRNA related to heterosis in the pink module and the dark red module. (**D**) The expression network of DEGs and DECs in the dark red module and the top 25 hub differentially expressed gens (DEGs) or hub DECs. (**E**) The expression network of DEGs and DECs in the pink module and the top 25 hub DEGs or hub DECs.

## Data Availability

The data that support the findings of this study are available from the corresponding author upon reasonable request.

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
