# Peer review of "Identification and Functional Prediction of CircRNAs in Leaves of F1 Hybrid Poplars with Different Growth Potential and Their Parents"

_ijms, 2023, doi:10.3390/ijms24032284_

Round 1

Reviewer 1 Report

The manuscript found that circRNAs may play a role in regulating the heterosis of poplar, especially the non-additive expression of circRNAs and SPE circRNAs. Through the correlation analysis of circRNAs expression and parental gene expression, it was found that circRNAs might regulate the heterosis of poplar by regulating the expression of their parental genes; Further, WGCNA was performed on circRNAs expression and protein coding gene expression, and key circRNAs and key biological processes involved in regulating poplar heterosis were screened. This paper studies the formation of poplar heterosis from the perspective of circRNAs, which can provide a reference for the molecular mechanism of the formation of woody plant heterosis.

It was my pleasure to review the manuscript, in which the authors had carried out a large number of measurements and analyses. However, there are still some problems in this manuscript. I suggest the manuscript may be accepted for publication after moderate revisions.

1.      Line 294-295, “F1 hybrid specifically expressed” means that circRNAs are not expressed in parents, but only expressed in F1 hybrids?

2.      Line 314-318, I didn't understand the relationship between 25, 34 and 61?

3.      Line 347-349, I didn't understand this sentence, please explain it in detail.

4.      Line 354-357, What does “92.2–100.0%” mean? all miRNAs targeted by circRNAs?

5.      Line 465-467, please explain the meaning of this sentence. I can't understand it accurately.

6.      Line 522-524, “which is consistent with previous research” means that the previous two conclusions are consistent?

7.      Please modify the annotation in Figure 6 to make the full text consistent, for example, there are “common nonadditive circRNAs in high growth potential F1 hybrids”.

8.      Line 785, in “MPV ((Y3MP+Y3FP)/2)”, do Y3MP and Y3FP refer to the expression of male parent and female parent respectively? Please specify.

9.      Explain the significance of Figure S24, which is not mentioned in the article.

10.  Please add a conclusion to this article.

11.  Please check the punctuation and space of the full text. Some of them have errors.

Author Response

Point 1: Line 294-295, “F1 hybrid specifically expressed” means that circRNAs are not expressed in parents, but only expressed in F1 hybrids?

Response 1: Yes, " F1 hybrid specifically expressed " means that circRNAs are not expressed in the parent, but only expressed in the F1 hybrids. And we also explained it in the notes of Figure 4C and D (Line 292-295) and in Line 298-299 of the text with red marked.  

Point 2: Line 314-318, I didn't understand the relationship between 25, 34 and 61?

Response 2: 61 is co-different expression circRNAs in the Y3L3 vs. Y3H1, Y3L3 vs. Y3H2, Y3L4 vs.Y3H1, and Y3L4 vs. Y3H2 (Figure 5A), from which, there were 34 up-regulated, 25 down-regulated in all four groups, and the other two are up-regulated or down-regulated in the four control groups. In order to facilitate understanding, we also modified it in the article (Line 319-321) with red marked.

Point 3: Line 347-349, I didn't understand this sentence, please explain it in detail.

Response 3: In this artricle, We identified 2059 circRNAs and 295 miRNAs that have targeted binding relationships. And among which, 544 circRNAs have only one miRNA-binding site, and 1515 circRNAs have multiple miRNA-binding sites (2-184). We also modified it in the article (Line 349-352) with red marked.

Point 4: Line 354-357, What does “92.2–100.0%” mean? all miRNAs targeted by circRNAs?

Response 4: "92.2 – 100.0%" is the proportion of miRNAs targeted by all identified circRNAs, which we have described in detail in Line 360-361 with red marked..

Point 5: Line 465-467, please explain the meaning of this sentence. I can't understand it accurately.

Response 5: After calculated the correlation between each module and genotype, we found that the correlation between the F1 hybrids with the same growth potential and each module was more similar, compared with F1 hybrids with different growth potential (Figure 7B). Such as , the pink module was significant positively correlated (0.52) in the high growth potential offspring (Y3H1 and Y3H2), and positively correlated (0.34) in the female parent, while negatively correlated (0.27 to 0.39) in the male parent and low growth potential offspring (Y3L3 and Y3L4), which indicating that DEGs and DECs in the pink module might positively regulate the high growth potential of poplar (Figure 7B). Otherwise, the dark red module was significantly positively correlated (0.55 to 0.70) in low growth potential offspring, but negatively correlated (0.22 to 0.30) in the other genotypes, suggesting that DEGs and DECs in the dark red module might negatively regulate the growth potential of poplar (Figure 7B). we also revised the description in the article (lines 465-477) with red marked.

Point 6: Line 522-524, “which is consistent with previous research” means that the previous two conclusions are consistent?

Response 6: “which is consistent with previous research” means that “The sequence length of the identified circRNAs is mainly 100~700 bp” and “most of the circRNAs were expressed at a low level” are consistent with previous research of Gossypium hirsutum and poplar. we revised it in the article(Line 528-531) with red marked.

Point 7: Please modify the annotation in Figure 6 to make the full text consistent, for example, there are “common nonadditive circRNAs in high growth potential F1 hybrids”.

Response 7: We have checked and modified it in Figure 6, mainly including modifying the notes in Figure 6B (marked in red, Line 380-382) and labels in Figure 6B and D (In the picture).

Point 8: Line 785, in “MPV ((Y3MP+Y3FP)/2)”, do Y3MP and Y3FP refer to the expression of male parent and female parent respectively? Please specify.

Response 8: Y3MP and Y3FP refer to the expression of circRNAs of male parent and female parent respectively. We change “MPV ((Y3MP+Y3FP)/2)” of the article to “MPV ((Y3MP+Y3FP)/2) (Y3MP was expression of circRNAs in male parent, Y3FP was expression of circRNAs in female parent)”. we revised it in the article(Line 781-783) with red marked.

Point 9: Explain the significance of Figure S24, which is not mentioned in the article.

Response 9: We are sorry for make this mistakes, Table S24 is the sequence information of all circRNAs we identified. And we added the sequence information of 7 hub circRNAs in table S23. So we deleted Table S24.

Point 10: Please add a conclusion to this article.

Response 10: According to suggestion, we have added conclusions in the article (Line 827-843) with red marked.

Point 11: Please check the punctuation and space of the full text. Some of them have errors.

Response 11: We have checked and modified the punctuation marks, spaces and other formats of the full text carefully.

Thank you very much for your review and suggestions on my manuscript. I have revised your suggestions one by one. Your suggestions have benefited me a lot. Once again, I would like to express my gratitude.

Reviewer 2 Report

Dear Authors,

The study used the leaves of F1 hybrids with high growth potential (Y3H1, Y3H2, and Y3H3), F1 hybrids with low growth potential (Y3L3 and Y3L4), and their parents of 3-year-old Populus deltoides for circRNA sequencing. The manuscript found that the non-additive expression of circRNAs and SPE circRNAs were involved in regulating the heterosis of poplar, and DECs in F1 hybrids with different growth potentials were involved in regulating the growth potential of poplar. The authors studied the role of circRNAs in poplar heterosis for the first time, and identified the key circRNAs and key biological processes regulating poplar heterosis, which can provide a reference for revealing the molecular mechanism of woody plant heterosis formation from a new perspective.

It was my pleasure to review the above-titled manuscript, in which the authors had carried out a large number of measurements and analyses. However, there are still some issues that need to be addressed.

1.      five F1 Hybrid Poplars with different growth potential and Their Parents were used in this tmanuscript , but the growth of them is not very clearly. Only in the material method (Line 748~750) introduced briefly, and it is recommended to supplement growth values to make the results more reliable.

2.      The authors identified several circRNAs that positively or negatively regulate poplar heterosis. But it is not clear how many circRNAs were identified in this paper, and what their predicted functions are?

3.      line 131, It is necessary to briefly introduce the relevant contents of reference 10.

4.      Line 171, please check if the note in figure 2C is correct.

5.      Line 118, The title of 2.2 is too broad, and it is recommended to modify it.

6.      Line 211-212, it is suggested to reconsider the title of 2.3.

7.      Line 317-318, please explain this passage in detail. The original text is confusing.

8.      Figure 6B and D, the annotation in the picture should be exactly the same as that in the article.

9.      Line 465-467, please check the accuracy of this sentence.

10.  Figure 7 is of poor quality (especially Fig 7B, 7D, 7E), which need to be redrawn. The annotation of 7C in the picture also needs to be consistent with that in the article.

11.  Check the full-text space and parentheses, for example, there are an error in “novel _circ_001330” in Line 504, “(BLP))” in Line 792.

12.  Figure S5 and S6, It is suggested to modify the GO enrichment analysis to a picture with p value.

13.  Figure S24It does not exist in the article. It is recommended to add the sequence information of important circRNAs in related tables.

14.  Most of the results are again repeated in discussion, discussion part needs to be strengthened. For example, Line 744-755, the last paragraph in discussion is suggested to be modified and added to the conclusion.

15.  In this article, there are two different screening thresholds for GO and KEGG enrichment analysis, and it is recommended to mark them in detail in each analysis.

16.  There were some writing errors about Table 20, Table 21, Table 22. The author didn't introduce Table S23 and Table S24 in the article. Please add.

17.  line 720 (the female parent was P. deltoides cl. '10/17'), which is different from the description of the female parent in line 74, please confirm.

18. The format of the references needs to be modified consistently.

Author Response

Point 1: five F1 Hybrid Poplars with different growth potential and Their Parents were used in this tmanuscript , but the growth of them is not very clearly. Only in the material method (Line 748~750) introduced briefly, and it is recommended to supplement growth values to make the results more reliable.

Response 1: Thanks for your advice, At the time of sample collection in this study, the average tree height of Y3H1、Y3H2 and Y3H3 was 8.60 m (8.69 m, 8.51 m, 8.59 m), and the average DBH of them was 6.20 cm (6.55 cm, 5.89 cm, 6.03 cm), which exhibited heterosis over higher parent for tree height (12.27% for Y3H1, 9.95% for Y3H2, and 10.98% for Y3H3) and DBH (12.93% for Y3H1, 1.55% for Y3H2, and 3.97% for Y3H3). On the other hand, average tree height of Y3L3 and Y3L4 was 7.30m (7.18 m, 7.50m) and the average DBH of them was 4.77cm (4.63 cm, 4.67 cm), which possessed negative heterosis over higher parent for tree height (-7.24% for L3 and -3.10% for L4) and DBH (-20.17% for Y3L3 and -19.48% for Y3L4). We also modified it in the article (Line 739-747) with red marked.

Point 2: The authors identified several circRNAs that positively or negatively regulate poplar heterosis. But it is not clear how many circRNAs were identified in this paper, and what their predicted functions are?

Response 2: A total of 3722 circRNAs were identified in this paper, of which 7 hub circRNAs were sclected by WGCNA, and two modules were sleceted which was significantly positively correlated in the high growth potential offspring (pink module) or in low growth potential offspring(dark red module). And then Cytoscape software was used to visualize the co-expression network of the pink module and the dark red module. At last, two hub DECs (novel_circ_000368 and novel_circ_001540) were identified in pink module, and five hub DECs (novel_circ_001330, novel_circ_003614, novel_circ_001672, novel_circ_002310, and novel_circ_003239) were identified in dark red module (Table S23; Figure 7D,E). According to the GO and KEGG functional enrichment analysis of pink module and dark red module, we speculate that these seven hub circRNAs may regulate poplar heterosis through carbohydrate metabolism, amino acid metabolism, energy metabolism and material transport.However, However, due to the lack of research on circRNAs, the specific functions of the them are still unclear, their specific functions in poplar heterosis require further analysis and verification. We also modified it in the article (Line 507-515) with red marked.

Point 3: line 131, It is necessary to briefly introduce the relevant contents of reference 10.

Response 3: We have added a brief introduction to the relevant content of reference 10 in the article with red marked (Line 132-133).

Point 4: Line 171, please check if the note in figure 2C is correct.

Response 4: we are sorry for make some mistake in the annotations in figure 2C. and we have re-inserted a new figure 2 (only modified figure 2C, E and F) in the text.

Point 5: Line 118, The title of 2.2 is too broad, and it is recommended to modify it.

Response 5: We have modify it as youe advice, and changed the title of 2.2 to “Differential Expression of circRNAs in Parents and F1 Hybrids with Different Growth Potential”with red marked (Line 118-119).

Point 6: Line 211-212, it is suggested to reconsider the title of 2.3.

Response 6: After considering, we revised the title of 2.3 as “Different Expression of SPE,CoPE and SFE circRNAs between Parents and the F1 Hybrids” (Line 214) with red marked.

Point 7: Line 317-318, please explain this passage in detail. The original text is confusing.

Response 7: 61 is co-different expression circRNAs in the Y3L3 vs. Y3H1, Y3L3 vs. Y3H2, Y3L4 vs. Y3H1, and Y3L4 vs. Y3H2 (Figure 5A), and form which, there were 61 were co-DECs (Figure 5A). Among these 61 co-DECs, 34 were all upregulated and 25 were all downregulated, and the other two were upregulated or downregulated in the four comparative groups. we also modified it in the article (Line 319-321) with red marked.

Point 8: Figure 6B and D, the annotation in the picture should be exactly the same as that in the article.

Response 8: According the advice, we have checked and modified the annotations in Figure 6B and D, and reinserted Figure 6 to make the article completely consistent with the picture (lines 377)

Point 9: Line 465-467, please check the accuracy of this sentence.

Response 9: After calculated the correlation between each module and genotype, we found that the correlation between the F1 hybrids with the same growth potential and each module was more similar, compared with F1 hybrids with different growth potential (Figure 7B). Such as , the pink module was significant positively correlated (0.52) in the high growth potential offspring (Y3H1 and Y3H2), and positively correlated (0.34) in the female parent, while negatively correlated (0.27 to 0.39) in the male parent and low growth potential offspring (Y3L3 and Y3L4), which indicating that DEGs and DECs in the pink module might positively regulate the high growth potential of poplar (Figure 7B). Otherwise, the dark red module was significantly positively correlated (0.55 to 0.70) in low growth potential offspring, but negatively correlated (0.22 to 0.30) in the other genotypes, suggesting that DEGs and DECs in the dark red module might negatively regulate the growth potential of poplar (Figure 7B). we also revised the description in the article (lines 465-477) with red marked.

Point 10: Figure 7 is of poor quality (especially Fig 7B, 7D, 7E), which need to be redrawn. The annotation of 7C in the picture also needs to be consistent with that in the article.

Response 10: According to the advice, we have redrawn Figure 7 (especially Fig 7B, 7D and 7E). The label of Figure 7C has also been modified to be consistent with the article.

Point 11: Check the full-text space and parentheses, for example, there are an error in “novel _circ_001330” in Line 504, “(BLP))” in Line 792.

Response 11: We have checked and modified the punctuation marks, spaces and other formats of the full text carefully.

Point 12: Figure S5 and S6, It is suggested to modify the GO enrichment analysis to a picture with p value.

Response 12: we redraw the GO enrichment analysis in Figures S5 and S6, which shows the significance of go items with p value.

Point 13: Figure S24,It does not exist in the article. It is recommended to add the sequence information of important circRNAs in related tables.

Response 13: We are sorry for make this mistakes, Table S24 is the sequence information of all circRNAs we identified. And we added the sequence information of 7 hub circRNAs in table S23. So we deleted Table S24.

Point 14: Most of the results are again repeated in discussion, discussion part needs to be strengthened. For example, Line 744-755, the last paragraph in discussion is suggested to be modified and added to the conclusion.

Response 14: according to the advice, we revised the last paragraph of the discussion section and added the conclusion section in order to make the logic of the article more reasonable. The details can be viewed in the article with red marked (lines 827-843).

Point 15: In this article, there are two different screening thresholds for GO and KEGG enrichment analysis, and it is recommended to mark them in detail in each analysis.

Response 15: I have explained in detail the screening threshold of GO and KEGG enrichment analysis in the article(lines 803-806, lines 811-814, and lines 822-825).

Point 16: There were some writing errors about Table 20, Table 21, Table 22. The author didn't introduce Table S23 and Table S24 in the article. Please add.

Response 16: Thanks for advice, we have checked table S20, table S21 and table S22 and corrected the wrong writing; Table S23 shows the genes and circRNA members in the pink module and dark module, and the relevant contents can be added in the article (lines 501-507,2.8); Table S24 is the sequence information of all circRNAs we identified. we deleted table S24 and added the sequence information of hub circRNAs in table S23.

Point 17: line 720 (the female parent was P. deltoides cl. '10/17'), which is different from the description of the female parent in line 74, please confirm.

Response 17: In this study, the female parent is P. deltoides cv. ‘55/65’, and the male parent is P. deltoides cl. '10/17'. we modified it in the article (lines 73, and 720) with red marked.

Point 18: The format of the references needs to be modified consistently.

Response 18: Thanks for advice, we checked carefully, and revised the format of references according to the requirements of the journal.

Thank you very much for your review and suggestions of my manuscript, which has greatly benefited me. I hope everything goes well with you, and express my thanks again!

Reviewer 3 Report

As general comment the work is well written and designed with relevant results.

The authors touch upon very important issues about the Identification and Functional Prediction of CircRNAs in Leaves of F1 Hybrid Poplars.

This manuscript timely and I commend the authors for bringing in some new ideas and analysis.

This study is very interesting and conforms to the requirements of the International Journal of Molecular Sciences journal.

The issues discussed by the Authors are original.

The manuscript is very well written.

The format and grammar of the paper are correct.

The abstract and Introduction chapter are properly written.

Materials and method section is well described and correspond to the aim set out in the manuscript. 

The results are correctly described.

Figures are clear and understandable.

The discussion is correct.

The conclusions of the article are correct and consistent with the discussed issues.

The references are properly chosen and cited.

I recommend the publication of this manuscript in the International Journal of Molecular Sciences journal in present form.

Author Response

Dear Reviewer,

Hope you are doing well.

Thanks very much for your reviewing our manuscript (“Identification and Functional Prediction of CircRNAs in Leaves of F1 Hybrid Poplars with Different Growth Potential and Their Parents” (Manuscript ID: ijms-2155359)) . and its my honor you very much for your recognition of our research.

Kind regards,

Yours,

Changjun Ding
